# Emotion diffusion effect: Negative sentiment COVID-19 tweets of public organizations attract more responses from followers

**Haiyan Yu[1], Ching-Chi Yang[2]\*, Ping Yu[3], Ke Liu[1]**

**1** Center for Data and Decision Sciences, Chongqing University of Posts and Telecommunications, Chongqing, China, **2** Department of Mathematical Sciences, University of Memphis, Memphis, TN, United States of America, **3** School of Computing & Information Technology, University of Wollongong, Wollongong, NSW, Australia

\* cyang3@memphis.edu

**Data Availability Statement:** All relevant data are within the paper and its Supporting Information files.

## Abstract

Coronavirus disease 2019 (COVID-19) has triggered an enormous number of discussion topics on social media Twitter. It has an impact on the global health system and citizen responses to the pandemic. Multiple responses (replies, favorites, and retweets) reflect the followers' attitudes and emotions towards these tweets. Twitter data such as these have inspired substantial research interest in sentiment and social trend analyses. To date, studies on Twitter data have focused on the associational relationships between variables in a population. There is a need for further discovery of causality, such as the influence of sentiment polarity of tweet response on further discussion topics. These topics often reflect the human perception of COVID-19. This study addresses this exact topic. It aims to develop a new method to unveil the causal relationships between the sentiment polarity and responses in social media data. We employed sentiment polarity, i.e., positive or negative sentiment, as the treatment variable in this quasi-experimental study. The data is the tweets posted by nine authoritative public organizations in four countries and the World Health Organization from December 1, 2019, to May 10, 2020. Employing the inverse probability weighting model, we identified the treatment effect of sentiment polarity on the multiple responses of tweets. The topics with negative sentiment polarity on COVID-19 attracted significantly more replies (69±49) and favorites (688±677) than the positive tweets. However, no significant difference in the number of retweets was found between the negative and positive tweets. This study contributes a new method for social media analysis. It generates new insight into the influence of sentiment polarity of tweets about COVID-19 on tweet responses.

## 1. Introduction

Social media such as Twitter has become a popular platform for thousands of people to exchange their thoughts about current affairs in the form of tweets [1]. It has a significant,

**Funding:** The author(s) received no specific funding for this work.

**Competing interests:** The authors have declared that no competing interests exist.

global influence on public opinions about current affairs [2, 3]. As of December 14, 2020, Coronavirus Disease 2019 (COVID-19) has infected 72.17 million people and resulted in 1.61million global deaths, according to the Johns Hopkins University COVID-19 Tally [4]. The disease burden of the pandemic unavoidably impacts population health and well-being, health resource utilization, social dynamics, world economies, and health technology development. Stakeholders around the world (i.e., governments, non-profit organizations, and healthcare communities) have taken active actions to respond to the COVID-19 epidemic.

Unsurprisingly, the COVID-19 pandemic has been triggering massive short informal texts on Twitter, expressing public anxieties, worries, and a new byproduct of discrimination [5]. Therefore, understanding and drawing insights from the massive number of social media posts, i.e., Tweets, has never been more critical. This understanding will assist in designing public health programs or events on social media to correct misinformation and overcome the current fear and stigma about COVID-19 globally. For example, the World Health Organization (WHO) tweeted on February 28, 2020 "everyone should know the #COVID19 symptoms?. . . Most people will have mild disease and get better without needing any special care." This tweet provides the domain knowledge about COVID-19 symptoms to the public and attracts positive sentiment. On March 11, 2020, WHO also tweeted, "some countries are struggling with a lack of capacity (resources, resolve)". This tweet shows WHO's negative sentiment (i.e., concern) about these countries' capacity to prevent, monitor, and control COVID-19.

Sentiment analysis is also known as opinion or emotion artificial intelligence [6]. It refers to the use of natural language processing (NLP), text analysis, computational linguistics, and machine learning to systematically analyze people's written language [7]. This result of sentiment analysis is acquired people's opinions, sentiments, evaluations, attitudes, and emotions. Sentiment analysis has been widely applied to draw insights from social media posts. It aims to determine the polarity of a piece of text as positive, negative, or neutral. Methods for these analyses often include counting the number of positive and negative words to determine the document's sentiment polarity. These methods often employ a dictionary of words with an assigned sentiment value, such as the lexicon of AFINN in the tidytext package [8]. The terms in AFINN are assigned scores from -5 to 5, with a negative score indicating negative polarity and positive polarity.

A high number of responses indicates a favorable effect of disseminating the message of the original tweet; therefore, understanding responses to a tweet has been a major topic of sentiment analysis. For example, cross-cultural polarity on COVID-19 related tweets was examined with emotion detection methods [9]. To date, few studies have examined the impact of a tweet's sentiment polarity (SP) on its responses (i.e., replies, favorites, and retweets). This study aims to fill this gap. We investigated the influence of sentiment polarity on the responses to COVID-19 in Twitter and examine the hypothesis: negative sentiment on COVID-19 will attract more responses in terms of replies, favorites, and retweets than positive sentiment. We conducted a quasi-experimental study on the publicly available Tweet data published between December 1, 2019, to May 10, 2020. These data were extracted using the keyword "coronavirus" or "COVID." The research method used is an innovative, causality discovery approach instead of the often-used association relationship.

## 2. Literature review

Studies of tweet responses have attracted substantial attention because they represent the human information sharing process on social media. The importance of tweet responses increases with Twitter being reported to outperform the mainstream media in numerous ways

[5]. For example, Li et al. found noticeable spikes in Twitter usage during disasters and other large events, suggesting Twitter is increasingly used as a fast response news service to voice public sentiment [5]. Araujo et al. recommend companies and brands should use Twitter to increase content distribution range and endorsement level [10].

There are three types of Tweet responses: reply, favorite, and retweet. The reply is the simple reply of a reader to the original tweet. It allows the reader to respond to the original tweet. Favorite allows the reader to express approval of the tweet. Retweet allows the reader to start a new discussion topic prompted by the original tweet, with the target audience being the reader's (retweets') followers. Naveed N et al. found bad News travel fast through content-based analysis of retweets [11].

Tweet responses are widely used for public sentiment analysis to investigate social-psychological trends. For example, Bae et al. measured the positive or negative influence of popular users on their Twitter audiences [12]. Their results suggest that popular users' impact on social media is consistent with their fame in the real world. Hswen et al. identified the psychological characteristics of Twitter users who self-identified with autism spectrum disorder [13]. Teufel et al. investigated the impact of German Chancellor Angela Merkel on the public's psychological distress, behavior, and risk perception during COVID-19 [14]. Their finding suggests that not all world leaders use Twitter in response to COVID-19 pandemic affairs. Eichstaedt et al. found that capturing community psychosocial characteristics through social media is feasible, and these characteristics are strong markers of cardiovascular mortality at the community level [15]. Balakrishnan et al. used the machine learning approach to analyze Twitter users' psychological features and improve the detection of cyberbullying [16]. They identified the correlation between the (positive) sentiment score of tweets and their responses; however, their study did not analyze the treatment effect of sentiment polarity on the number of responses.

These research projects have all employed the sentiment polarity of tweets, labeled as positive (+1) or negative (-1), to identify social psychological trends in sentiment analysis [17]. There are two contradictory views about the impact of positive (or negative) sentiment polarity on tweet responses; one suggests that (positive) sentiment polarity enhances a tweet' responses in terms of replies, favorites, and retweets [18]; and these will further improve information sharing on social media. The opposing view suggests the other way around; the negative polarity leads to the above positive effects, i.e., increasing the number of replies and favorites, but without a significant effect on retweets [19]. This research aims to uncover the truth of these opposing views.

Sentiment score is commonly recorded as a positive number for positive sentiment and a negative number for negative sentiment. For example, the sentiment score of an entry *great* is (positive) 1.2, stating this entry has positive polarity with an evaluative score of 1.2 (see more details in [17]). The sentiment score of an entry *acceptable* is (positive) 0.1, suggesting that the entry *acceptable* has positive polarity 0.1, lower than that of the term *great* [17]. The sentiment triggered by the entry *victim* [16] is usually associated with negative emotions such as *anxiety* (-2) in tweets within the corpus AFINN [8]. However, it is still not clear how sentiment polarity impacts responses (replies, favorites, and retweets).

Experiments are difficult to conduct when researching social media data. The main reason lies in challenges in recruiting a large number of research participants (comparable to the normal population) and replicating the social reactions in laboratory settings. Quasi-experimental research provides a feasible and low-cost approach to identify the treatment effect of sentiment polarity [17–20]. Therefore, studies of the treatment effect of social media, i.e., Twitter, mainly implement a quasi-experimental study design to examine the causal effect of a Tweet's sentiment polarity. For example, using a fixed-effect model, Mousavi et al. addressed potential

endogeneity in the 111th U.S. senators' Twitter adoption decisions in 24 months [20]. They used the method of propensity score matching [21] and difference-in-differences approaches [22] to identify causal relationships of lawmakers' voting orientations. Semiparametric estimation can also be employed for inference about the treatment effect when treatment intake is likely not randomized [23].

The literature on covariates of tweet responses is mainly focused on discovering determinants for Tweet response rates [18]. Many factors, such as the number of tweets, average document (tweet) length, and the number of discovered events/topics, are applied as the covariates in the temporal mining of micro-blog texts [24]. For COVID-19 related studies, Pan et al. employed other external features (i.e., confirmed cases and deaths due to COVID-19) as the covariates for a public health intervention study [25]. Although there are some drawbacks (i.e., high signal-noise ratio) with the covariates from tweet features, the bias [26] could be quantified through the process of covariate balance with inverse probability weighting [21]. Terry investigated the critical differentiator for social media [27]. On choosing the study population of tweets, Terry chose professional journalists (or other paid content providers) and ordinary, common users. In Terry's terms, the content created by professional users or other paid content providers is different from that generated by common users without professional training, thus the former should be given higher weight.

The types of health topics discussed on Twitter were investigated in [28] and they found that tweets can both augment existing public health capabilities and enable new ones. Through the analysis of a unique Twitter dataset captured in the early stage of the current Ebola outbreak, the results provided insight into the intersection of social media and public health outbreak surveillance [29]. Their study suggests that Twitter mining is useful to inform public health education. Moreover, a systematic review also suggests that there is an essential need for an accurate and tested tool for sentiment analysis of tweets using a health care setting [30]. For example, the natural language processing approach and healthcare-specific corpus of manually annotated tweets were implemented to learn the sentiment from Texas Public Agencies' tweets and public engagement during the COVID-19 pandemic [31].

This study aims to investigate the impact of the sentiment polarity on the responses to COVID-19 related tweets. The response rate refers to the average number of responses to an organization's weekly tweets. It is calculated by dividing the total number of responses to an organization's weekly tweets by the number of weekly tweets published by this organization.

*Hypothesis 1*: For an organization's weekly tweets, there is no significant difference between the response rate of ***replies*** with a negative sentiment polarity and that with a positive sentiment polarity.

*Hypothesis 2*: For an organization's weekly tweets, there is no significant difference between the response rate of ***favorites*** with a negative sentiment polarity and that with a positive sentiment polarity.

*Hypothesis 3*: For an organization's weekly tweets, there is no significant difference between the response rate of ***retweets*** with a negative sentiment polarity and that with a positive sentiment polarity.

As the organizations' weekly tweets on COVID-19 are not neutral sentiment polarity but either being positive or negative (see Section 4.1), the hypothesis only compares negative and positive sentiments.

**Table 1. Statistics of the Twitter accounts of the nine studied public organizations.**

| ID | Entity | Org. | Location | Joined | Tweets1 | Following | Followers | Tweets2 |
|---|---|---|---|---|---|---|---|---|
| 1 | CDCgov | USCDC | US | May-10 | 26.6K | 267 | 2.6M | 441 |
| 2 | ChinaCDC | CNCDC | China | Jan-20 | 112 | 35 | 216 | 85 |
| 3 | healthgovau | AUDoH | Australian | Mar-10 | 15K | 135 | 79.6K | 457 |
| 4 | NHSEngland | UKNHS | UK | Apr-12 | 42.8K | 2379 | 415K | 313 |
| 5 | WHO | WHO | Inter-national | May-09 | 12.8K | 1744 | 74.5K | 2039 |
| 6 | australian | Australian | Australia | Oct-07 | 242.1K | 545 | 717.9K | 1084 |
| 7 | bbcworld | BBC | UK | Feb-07 | 296K | 70 | 25.3M | 1411 |
| 8 | ChinaDaily | CDaily | China | Nov-09 | 97K | 490 | 3.95M | 6394 |
| 9 | cnn | CNN | US | Feb-07 | 248K | 1107 | 42.1M | 7694 |

Notes: US: United States; UK: United Kingdom; NA: Missing; K denotes thousand; M denotes million.

WHO: World Health Organization; Org.: Organization.

Joined: Date of the organizational account joining the Twitter platform; May-10 means May 2010.

Tweets 1: Number of total tweets.

Tweets 2: Number of Tweets during the study period.

ChinaCDC: China CDC Weekly, Platform of Chinese Center for Disease Control and Prevention.

## 3. Models and statistical analysis

### 3.1 Study context and measurements

The population is the tweets collected in 24 weeks (from December 1, 2019 to May 10, 2020) from nine Twitter accounts: five government organizations and four major news agencies [32] (see Table 1). The government organizations are the Australian Government Department of Health (AUDoH with healthgovau as its Twitter account), Chinese Center for Disease Control and Prevention (CNCDC), National Health Service (UKNHS with NHS England as its Twitter account), CDC of the US (USCDC) and World Health Organization (WHO). The news agencies are The Australian (australian), China Daily (CDaily), BBC in the UK, and CNN in the US. These four agencies are the most popular, with a massive number of followers on the Twitter platform from the four countries. The reasons for including the news agencies along with government organizations lie in two aspects. First, news agencies were often employed in the literature to investigate the social media impact on an emergent event, e.g., the use of social media in the Westgate mall terror attack in Kenya [33]. Second, this paper employed the news agencies that are the representative media account on the Twitter platform from the four countries. These news agencies are one of the main communication channels of the stakeholders (i.e., governments, non-profit organizations, and healthcare communities that have taken active actions to respond to the COVID-19 epidemic) to the public.

We collected data about the weekly number of tweets from the five government organizations and four news agencies as distinguished by each Twitter account. One week is defined from Sunday 12 am to the next Saturday at 12 am. The overall study period is 24 weeks. Tweet responses (as dependent variables) include replies, favorites, and retweets (see Table 2). The unit of analysis is the weekly average number of replies (RP), average number of favorites (FV), and average number of retweets (RT) for weekly tweets of each of the nine organizations.

Because the treatment variable of interest is the sentiment polarity (SP), we divided the data into two groups: the positive group with the treatment factor SP = 1, the negative group with SP = 0.

Eight covariant variables are also employed in this study (see Table 2). Five covariates were adopted from Mousavi et al. [20] investigating the impact of Twitter adoption on lawmakers'

**Table 2. Variable definitions and measurements.**

| Variables | Definitions | Measurements |
|---|---|---|
| Dependent variable (multiple responses) | | |
| RP | Mean of replies | The average number of replies for an organization's weekly tweet. |
| FV | Mean of favorites | The average number of favorites for an organization's weekly tweet. |
| RT | Mean of retweets | The average number of retweets for an organization's weekly tweet. |
| Objective | | |
| DET | Differential effect of treatment | DET = mean of $Y$ (SP = 1,X)—mean of $Y$ (SP = 0,X) |
| *Control variable (Treatment)* | | |
| SP | Sentiment polarity | A dummy variable indicating the status of the sentiment score. 1 indicates the positive score, otherwise 0. |
| Covariates (X) | | |
| NT | Number of tweets | The number of an organization's weekly tweets. |
| LT | Length of Text | The total length of an organization's weekly tweets. |
| MF | Median frequency | Median frequency of terms in an organization's weekly tweets. |
| NI | Number of items | The total number of terms in an organization's weekly tweet. |
| HF | Highest frequency | Highest frequency of terms in an organization's weekly tweets. |
| CC | Confirmed cases | The number of weekly confirmed COVID-19 cases in an organization's registered country, and total number of worldwide cases for WHO. |
| ND | Number of deaths | The number of weekly deaths due to COVID-19 in an organization's registered country, and total number of worldwide deaths for WHO. |
| OG | Indicator of organizations | OG = 0 indicating the entity is a government organization; OG = 1 indicating a news agency. |

Notes: Unit of analysis for each variable is the weekly count data (from last Saturday to this Sunday) of a Twitter account.

Terms: words in a tweet.

voting orientation. Two covariant variables were identified from COVID-19 weekly statistics. The remaining covariate (OG) is a blocking factor for the data set, indicating whether the entity comes from the media agencies.

The former five covariates included the weekly number of tweets (NT) of an organization, length of text (LT) of the organization's weekly tweets, the median frequency (MF) of the weekly tweets, number of items /words (NI), and highest frequency (HF) of the weekly tweets. They can be extracted from tweet text mining [34].

The latter two covariates included the weekly confirmed COVID-19 cases (CC) and the weekly number of deaths (ND) caused by COVID-19. These data were collected from the Johns Hopkins COVID-19 Daily Tally: the weekly number of confirmed cases and the weekly number of deaths caused by COVID-2019 in the region of the nine authoritative public organizations to be studied and worldwide for WHO. For example, the confirmed cases and deaths in Australia are linked with the Twitter accounts of healthgovau and australian. The data in China are linked with China CDC Weekly and China Daily's Twitter accounts. Moreover, the worldwide confirmed cases and deaths are associated with the Twitter accounts of WHO. All the covariates are stock variables representing the samples' historical data at the data acquisition time.

## 3.2 Study design

Randomization inference with the quasi-experimental design is the approach taken for causality discovery. A quasi-experimental design will allow the investigator to control the assignment

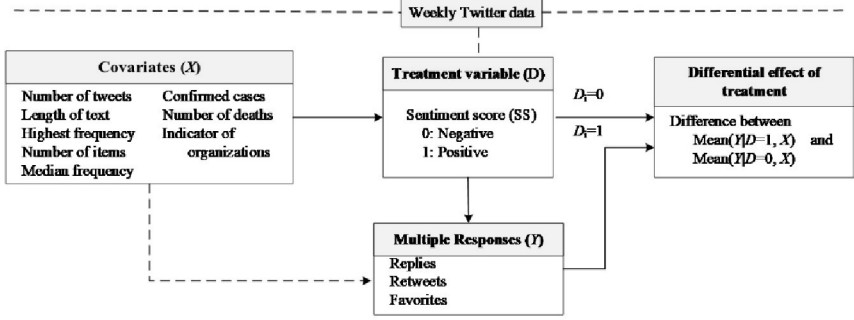

**Fig 1. The study framework.**

to the treatment condition, using certain criteria (i.e., covariate balance diagnosis) other than random assignment. Therefore, we conducted a quasi-experimental study to answer the research question (i.e., what is the treatment effect of sentiment polarity on tweet responses).

Sentiment polarity (SP) is inferred from the Tweet responses (replies, favorites, and retweets) and used as the treatment variable, with two levels (positive or negative). Thus, the population (weekly tweets) is divided into negative and positive groups by this treatment variable. This quasi-experimental study is aimed at estimating the causal impact of the intervention (SP) on the target population without random assignment. Fig 1 presents the research framework.

The differential effect of treatment (DET) refers to the difference between the mean of the responses (i.e., weekly RP, RT, and FV) between the treatment (SP = 1) and control (SP = 0) group conditioned on their balanced covariates. The covariates (i.e., the number of tweets, average document (tweet) length, the number of discovered events/topics) are applied to predict the balancing score of each weekly tweet with logistic regression. This score is further used in the inverse probability weighting model, which produces the covariate-adjusted means for the treatment effect. This model reduces the bias caused by the covariates.

The sentiment polarity is measured based on the sentiment score (SS) [17]. For a tweet, it is calculated weekly.

$$SP = 1_{(\sum_{w \in Twitter_i} Score(w) > 0)}, \tag{1}$$

where $Score(w) = log_2 \frac{freq(w,positive)*freq(negative)}{freq(w,negative)*freq(positive)}$, $freq(w, positive)$ is the number of times a term $w$ occurs in positive tweets, $freq(positive)$ is the total number of each term in positive tweets; similar terminologies are defined for $freq(w, negative)$ and $freq(negative)$ for the negative tweets. $SP$ is an indicator function, representing the positive polarity of responses. As introduced above, $SP$ is a binary variable (see Table 2).

### 3.3 Data collection

The study organizations were chosen following the recommendation of [32]. Although published research [35] set up a comprehensive timeline of the spreading of COVID-19, we considered it useful to start initial data collection one week before the first reported case in Wuhan to set up the pre-COVID benchmark. The data collection was concluded at the week including May 10, 2020, to avoid the impact of the Black Lives Matter (BLM) movement [36] (see Fig 2). The COVID-19 data were collected from the Johns Hopkins University COVID-19 Tally [4]. Tweets were collected in three time periods.

A. No. of COVID-19 confirmed case

B. No. of COVID-19 deaths

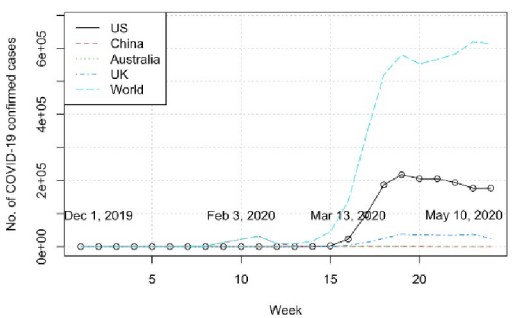
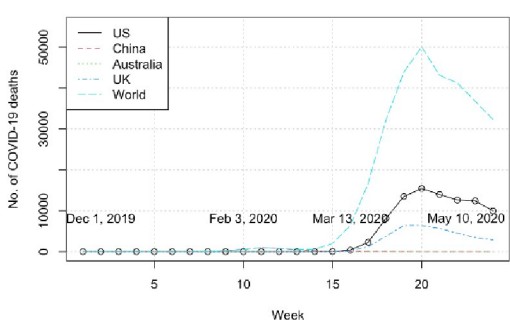

**Fig 2. Weekly number of COVID-19 confirmed cases and deaths.**

Stage 1. Commenced December 1, 2019, to February 2, 2020, one week before the first reported case in Wuhan, and ended on February 2, 2020, when the centralized treatment and isolation strategy was established in China.

Stage 2. February 3, 2020, to March 12, 2020, the period marked by the inception of the isolation strategy in China and the travel ban to the US from Europe, when the WHO declared the COVID-19 outbreak a global pandemic.

Stage 3. March 13, 2020, to May 10, 2020, a period marked by the start of the travel ban in the US and concluded on May 10, 2020, the inception of the 'Black Lives Matter' movement.

Using web crawler technology with Twitter API, the text data from Twitter were collected on May 20, 2020. The libraries, *tm*, *NLP*, and *plyr* were incorporated to conduct text mining in R (RStudioTeam, 2016) [37].

The nine organizations posted a total of 56,557 tweets during the 24 week study period. The data was filtered with the terms that were referenced from Wikipedia [38]: "Coronavirus disease 2019", "COVID-19", "Coronavirus", "Corona", "COVID", "2019-nCoV acute respiratory disease", "Novel coronavirus pneumonia", "Severe pneumonia with novel pathogens", "Wuhan Acute Respiratory Syndrome (WARS)", and "SARS-CoV-2".

A total of 19,918 tweets remained in the data set after filtering with these terms. The tweets were aggregated into weekly records, i.e., one organization only has one tweet record per week. Thus, the resulting aggregated data has 216 (= 9*24) rows.

## 3.4 Analysis models

The data was first fitted with the balancing score model that takes into account the treatment and covariates (*X*), as shown in Eq (2). Then the model was fitted with responses (*Y*) for the treatment effect of emotion diffusion (sentiment polarity), as shown in Eq (3) and (4). All variables are defined in Table 2.

$$p_i(SP_i = 1|X) = \text{logit}\left(\begin{array}{c} \beta_0 + \beta_1 NT_i + \beta_2 LT_i + \beta_3 MF_i + \beta_4 NI_i \\ + \beta_5 HF_i + \beta_6 CC_i + \beta_7 ND_i + \beta_8 OG_i \end{array}\right) \tag{2}$$

$$DET = mean(Y|SP = 1, X) - mean(Y|SP = 0, X) \tag{3}$$

$$DET' = \frac{1}{n}\sum_{i}\frac{[Y_i \cdot 1_{(SP=1)}]}{p_i} - \frac{1}{n}\sum_{i}\frac{[Y_i \cdot 1_{(SP=0)}]}{(1-p_i)} \qquad (4)$$

where $\beta_0$ is the constant term, and $\beta_1,\ldots,\beta_8$ are the coefficients of the covariates ($X$); $p_i$ is the balancing score of the treatment on the covariates. It is the function of the observed covariates such that the conditional distribution of covariates (given $p_i$) is the same for the unit (one of the 216 samples) in the two groups (SP = 0, 1). In the treatment (SP = 1) group, the weight of each unit is $1/p_i$; in the control (SP = 0) group, the weight of each unit is $1/(1-p_i)$. For example, when the balancing score of a unit is 0.4, the weight is 2.5 for the unit in the treatment group with $p_i = 0.4$; and the weight is 1.6667 (= 1/0.6) for the unit in the control group with $p_i = 0.4$. When the balancing score of a unit is 0.5, the weights are 2 for the units in both the treatment and the control group with $p_i = 0.5$.

Eq (2) presents a logistic regression model that captures the covariates' influence on the treatment effect. Eq (3) expresses the differential effect of treatment, which is the difference in emotion diffusion between the negative and positive sentiment groups. Each term in Eq (4) is a mean response adjusted by covariates. Eq (4) is an inverse probability weighting (IPW) estimator of the treatment effect inference model, which is built with the IPW method [21].

## 4. Results

### 4.1 Descriptive results

In the original 216 samples (see Table 3), the sentiment score ranged from -232 to 900, with a mean of 24.21 and a median of zero. There were fourteen rows with zero scores because most organizations did not release tweets related to COVID-19 topics in the first two weeks. To fit the regression model with the two groups (positive and negative), these samples were removed; thus, only 198 samples were used to fit the models (N = 198). Meanwhile, the average number of replies (RP) ranged from 0 to 303.6, with a mean of 37.65 and a median of 10. The average number of retweets (RT) ranged from 0 to 2667, with a mean of 173.9 and a median of 31.08. The average number of favorites (FV) ranged from 0 to 4305, with a mean of 363.2 and a median of 60.61.

**Table 3. Statistics of the Twitter accounts of nine authoritative public organizations.**

| Variable | Name | Min. | 1Q | Median | Mean | 3Q | Max. |
|---|---|---|---|---|---|---|---|
| SS | Sentiment score | -232 | -9.25 | 0 | 24.2 | 15.5 | 900 |
| RP | Mean of replies | 0 | 0 | 10 | 37.7 | 54.8 | 303 |
| FV | Mean of favorites | 0 | 0 | 60.6 | 363 | 581 | 4305 |
| RT | Mean of retweets | 0 | 0.21 | 31.1 | 174 | 258 | 2667 |
| NT | Number of tweets | 0 | 1 | 24.5 | 92.2 | 108 | 757 |
| LT | Length of Text | 0 | 3.75 | 459.5 | 1576 | 1407 | 12639 |
| MF | Median frequency | 0 | 0.75 | 1 | 1.03 | 1 | 7 |
| NI | Number of items | 0 | 3.75 | 195.5 | 567 | 708 | 3552 |
| HF | Highest frequency | 0 | 0.75 | 28 | 83.4 | 78.5 | 714 |
| CC | Confirmed cases | 0 | 0 | 92 | 38424 | 10084 | 619634 |
| ND | Number of deaths | 0 | 0 | 1.5 | 2602 | 428 | 50045 |
| OG | Indicator of organizations | 0 | 1 | 1 | 0.587 | 1 | 1 |

Note: Q: quarter. The aggregated data has 216 (= 9*24) rows. Each row represent the statistics of an organization's weekly tweets or cases. The 24 weeks are from Decision 1, 2019 to May 10, 2020.

The weekly number of confirmed COVID-19 cases (see Fig 2) showed two trends: a peak in the middle of February in China, and a sharp growth after March 13 in the US and UK. The weekly number of COVID-19 related deaths increased after March 13 in the US and UK. However, these numbers in Australia remained low in comparison with the other regions.

In the positive tweet group, except for the CDCs of the US and China (USCDC and CNCDC), the sentiment scores of the tweets from the other seven organizations significantly increased after March 13, 2020 (see Fig 3). Twenty-one out of the 24 weekly tweets published by the USCDC had positive sentiment scores, except three negative weekly tweets in March and May (see Fig 3A). The tweets from the CNCDC had negative but close to zero scores (Fig 3C). The tweets from UKNHS and the Australian Government Department of Health (AUDoH) showed a similar, increasingly positive trend. The WHO's tweets had negative polarity in February and May, but positive in the rest of the study period. There was a peak in the number of tweets from the four news agencies in February (see Fig 3B). China Daily published positive sentiment tweets in most of the study period except the week of December 8 to 14, 2019 (negative). The BBC and The Australian had negative tweets all of the time. Tweets from CNN were mostly negative but were positive for two time periods: from February 9 to 19, and from March 1 to March 14, 2020. On the response rates, the USCDC received the highest number of responses, followed by WHO, UKNHS, and AUDoH (Fig 3E). The CNCDC received the least (almost zero response). Of the four news agencies, the tweets of CNN attracted the largest response rate, followed by the BBC (Fig 3F). China Daily and The Australian attracted the least number of responses.

With positive tweets, the USCDC attracted the largest number of replies, favorites, and retweets, followed by the WHO (see Fig 4A, 4C and 4E). Ranked third and fourth were the UKNHS and AUDoH. The CNCDC received the least response. The response rate to the negative tweets (see Fig 4B, 4D and 4F) of the five government organizations showed a similar trend to those for the positive tweets. All the tweets from the AUDoH were positive but did not receive any response. The BBC's negative tweets attracted a larger number of replies than those in the positive group.

## 4.2 Distributions of balancing scores and weights

The results of the balancing score modeling (see Eq (2) in section 3.4) are presented in Fig 5. Each sample case's weight is obtained with the inverse probability (balancing score) for the positive and negative tweet groups.

The balancing scores did not overlap between the two groups (Fig 5A). This result suggests that the covariates of the two groups lack balance. The average balancing score of the negative tweets (group) is less than 0.5 and has a large variance. The average balancing score of the positive tweets (group) is larger than 0.5 and has a small variance.

The lack of covariate balance often leads to bias in inference about treatment effect. The inverse probability weight was used for adjusting the mean response's estimation with the covariate-adapted mean. This has resulted in the negative group having a wide range of weights from 1 to 10, while the positive group has a narrow range of weights mainly from 1 to 2.5. To reduce the impact of covariate imbalance, samples' trimming is set at the thresholds of 0.3 and 0.7 of the balancing score. The cut-offs of the weights are set as 1.43 and 3.33 (Fig 5B).

## 4.3 Differential effects

Using the model (4) in section 3.4, the differential effect of treatment is obtained with the covariate-adapted mean of the outcomes (average number of responses to the weekly tweets). It implies the difference in response between the negative and positive tweet groups. The larger

A. Sentiment score of five government organizations

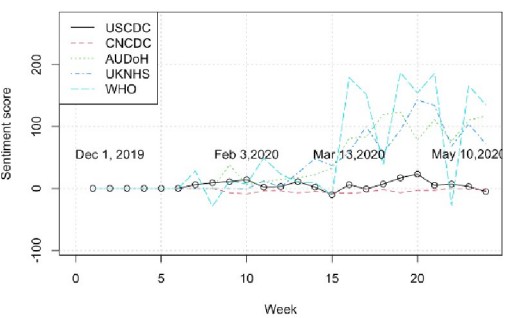

B. Sentiment score of four news agents

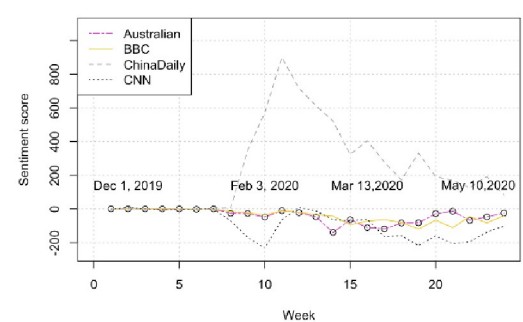

C. Distributions of sentiment scores

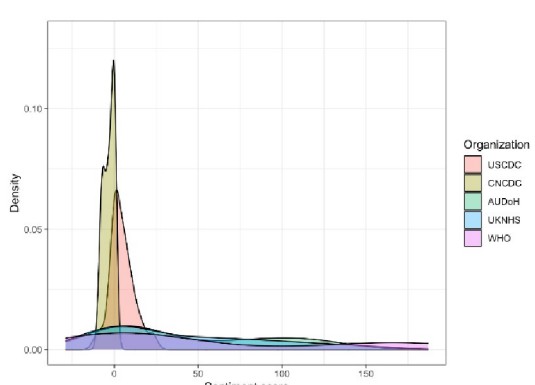

D. Distributions of sentiment scores

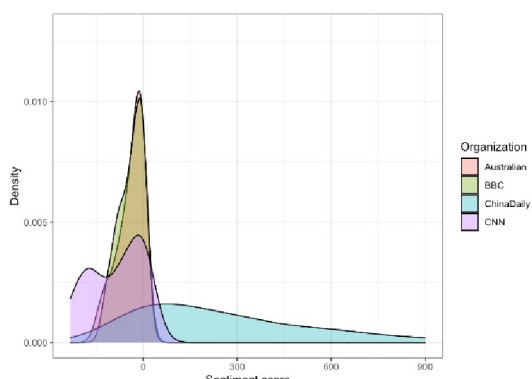

E. Scatter plot of responses to five government organizations' tweets

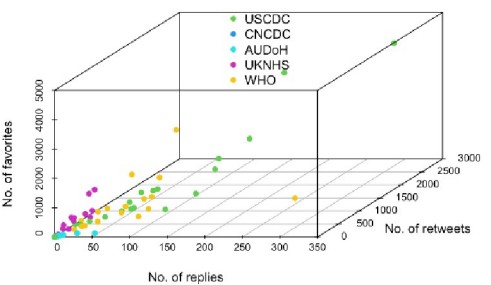

F. Scatter plot of responses to four news agents' tweets

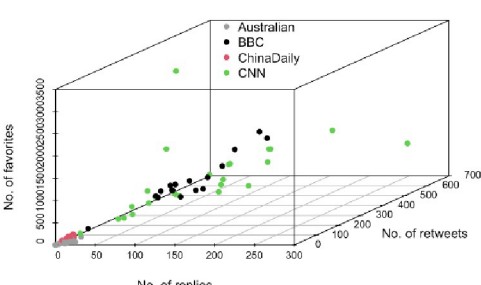

**Fig 3. Sentiment score, their density and scatter plots of responses to nine organizations' weekly tweets.**

A. Number of replies to positive tweets

B. Number of replies to negative tweets

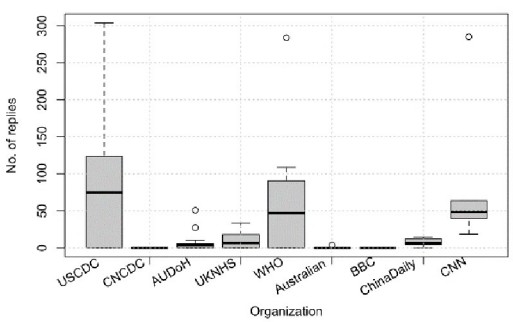
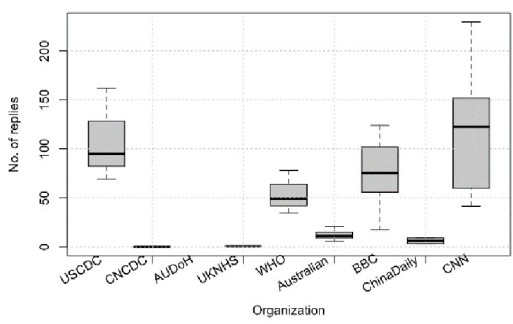

C. Number of favorites to positive tweets

D. Number of favorites to negative tweets

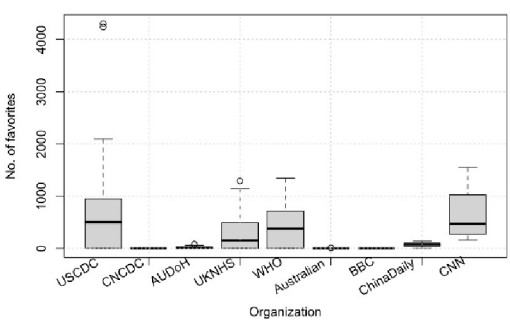
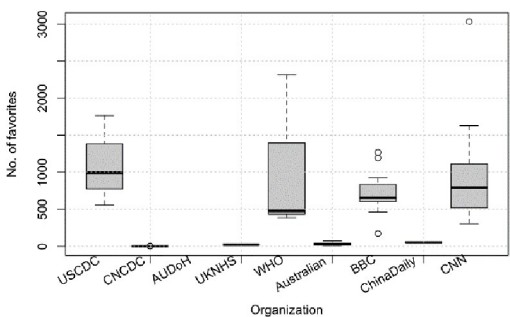

E. Number of retweets to positive tweets

F. Number of retweets to negative tweets

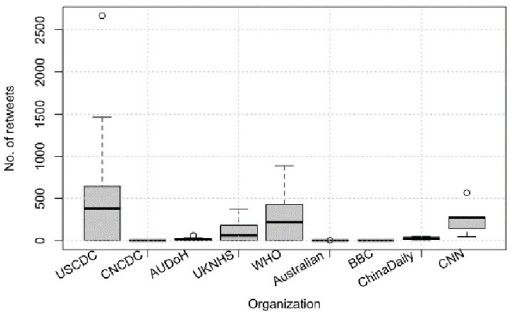
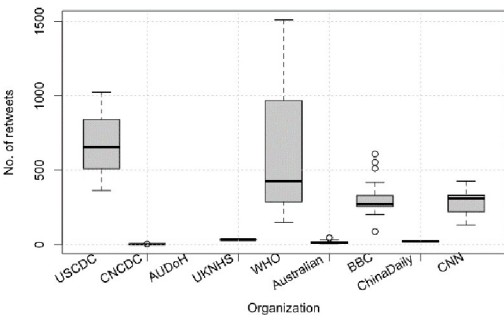

**Fig 4. Number of responses (replies, favorites, retweets) to positive and negative tweets from the five government organizations and four news agencies.**

A. Density of balancing score

B. Density of case weights with inverse probability

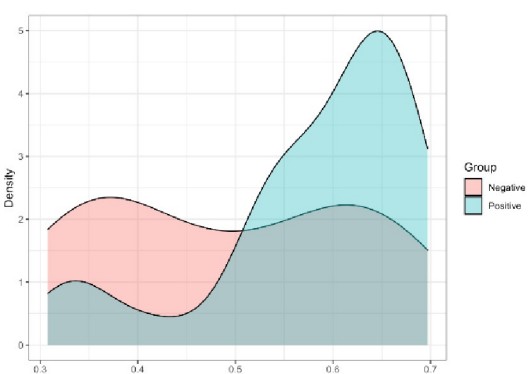
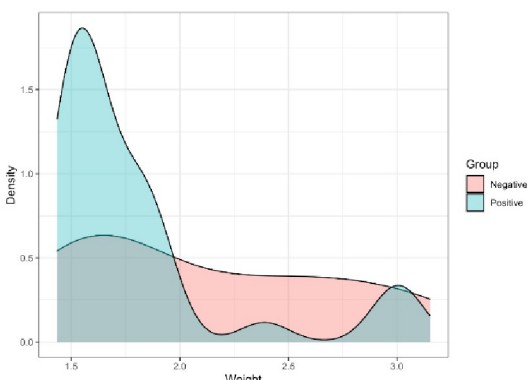

**Fig 5. Distributions of the balancing score and case weights.**

the absolute value of the treatment effect, the larger the size of the sentiment polarity's impact on the response to the weekly tweets. The results (see Fig 6) with the weighting method are compared with those of the unadjusted for the three data sets: the pooled data of the government and news agency, the government organization data, and the news agency data. Tweets with negative sentiment polarity on COVID-19 attracted more replies and favorites than the positive ones. However, no significant difference was found between the retweets of these negative and positive tweets.

For the replies (see Fig 6A), the differential effects of treatment with the weighting method are -69.95 (±49.01), -96.48(±138.52), and -54.78 (±69.25) with the pooled, government organization and news agency data. For the pooled data, the results suggest that the replies to the organizations are more (69.95±49.01) for the negative topics on COVID-19 than the positive ones. By contrast, the treatment effects with the unadjusted method are smaller in absolute value than those of the weighting method. The differential effects of treatment with the unadjusted method are -25.86 (±15.59), 10.73 (±22.00), and -50.59 (±21.30) with the three data sets. The number of replies is much larger for the negative topics on COVID-19 than for the positive topics for tweets from both government and news agency organizations. The treatment effect of negative polarity on the replies of weekly tweets has a larger impact (69.95>25.86) than those with the unadjusted method.

For the favorites (Fig 6B), the treatment effects with the weighting method are -688.45 (±677.05), -1594.09 (±2351.73), and-371.64 (±499.27) with the pooled, the government-organizational and news agency data sets. The result of the pooled tweets' favorites suggests that the negative tweets win a larger (688.46±677.05) number of favorites than the positive ones. By contrast, the treatment effects with the unadjusted method are much smaller in the absolute value of the treatment effect. Their treatment effects of sentiment polarity are -185.67 (±166.75), 63.78 (±306.73), and -426.58 (±174.79) with the three data sets. For the government-organizational data set, the treatment effect of sentiment polarity on the favorites has a standard deviation too wide to derive a scientific result. The treatment effect of negative polarity on the favorites of weekly tweets also has a larger impact (688.45>185.67) than those with the unadjusted method.

For the retweets (Fig 6C), the treatment effects with the weighting method are -325.20 (±416.25), -1027.15 (±1493.90), and -133.80 (±156.45) with the pooled, the government-organizational and news agency data sets, respectively. By contrast, the treatment effects with the

A. Differential effect of treatment on replies

B. Differential effect of treatment on favorites

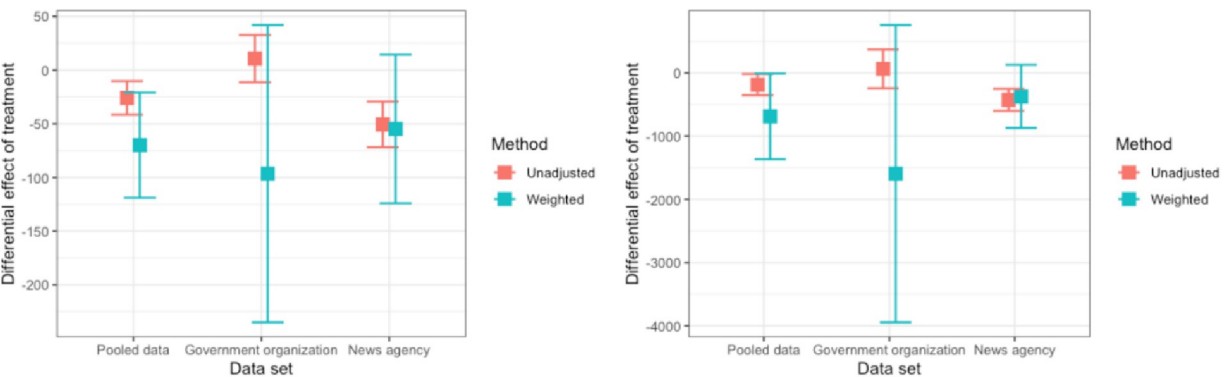

C. Differential effect of treatment on retweets

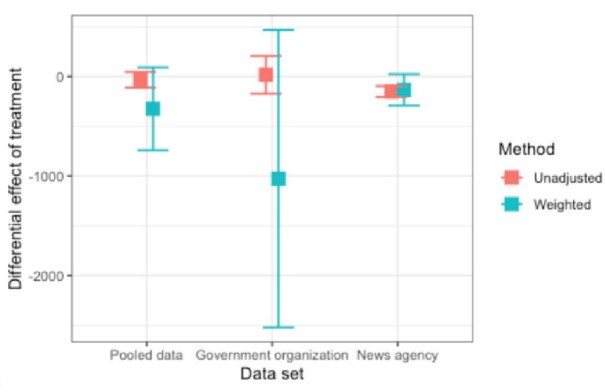

**Fig 6. The differential effect of treatment on the responses (replies, favorites, and retweets) before and after justification for the covariates.**
The larger the absolute value of the treatment effect is, the larger the size of sentiment polarity's impact is on the responses of the weekly tweets.
The treatment effects represent the mean difference in responses to the tweets with negative sentiment and those responses to the tweets with
positive sentiment from the five government organizations and four news agencies. pooled data = government organizations + news agencies.

unadjusted method are -33.61 (±79.47), 16.19 (±189.95), and -151.47 (±55.17) with these three
data sets. The treatment effects on the retweets have a standard deviation that is too wide,
which is larger than the values of the mean. Thus, the results suggest that there is no difference
in the sentiment polarity's impact on the responses of retweets to the weekly tweets.

In summary, the results (in Fig 6) are significant for replies (p = 0.001 with unadjusted and
p = 0.006 with weighted) and favorites (p = 0.029 with unadjusted and p = 0.046 with
weighted). Although the p-value of retweets reduced from 0.405 to 0.124, it is still not signifi-
cant. The treatment effect values with the weighting method are lower than the unadjusted val-
ues. Their standard deviations with the weighting are wider than the unadjusted values.

## 4.4 Stability of the effect

Statistical stability can effectively aid the pursuit of interpretable and reliable scientific models.
The stability of statistical results is relative to reasonable perturbations to data and the model
used. The following results with bootstrap (parameter n = 500) show the stability of the differ-
ential effect of sentiment polarity on the responses to the weekly tweets.

### A. Linear regression of replies with sentiment score

### B. Linear regression of favorites with sentiment score

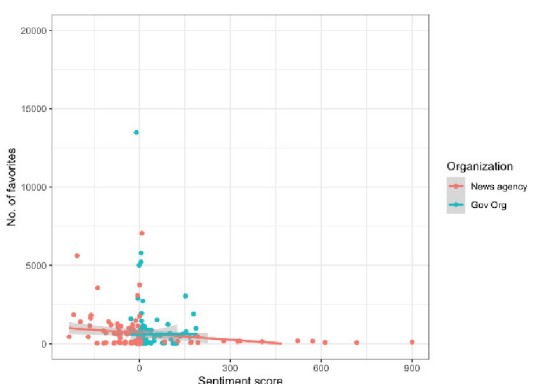

### C. Linear regression of retweets with sentiment score

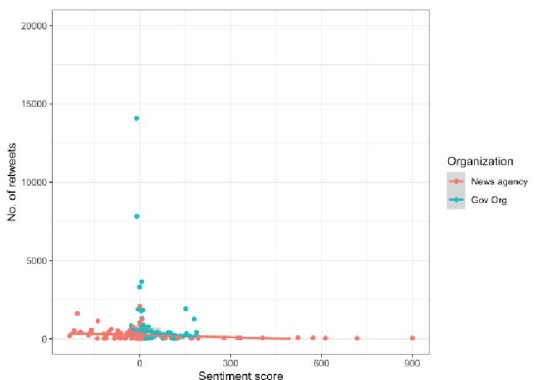

**Fig 7. Stability of the differential effect of treatment (linear regression of sentiment score on multiple responses).** Gov Org: Government organization.

The points in Fig 7 denote the responses with the weighting method. The lines are fitted with the linear models. The results show that the slopes of the fitting lines are negative, consistent with the treatment effects (provided in Section 4.3).

T-test result suggests that the differences in responses to the tweets with negative and positive sentiments are significant for replies ($p = 0.001$) and favorites ($p = 0.029$) but not significant for retweets ($p = 0.405$). Thus, we can reject the null hypothesis (on replies and favorites) and accept the corresponding alternative hypothesis (the negative sentiment polarity impacts the response rates of replies and favorites). The results suggest that there are true differences in RP and FV between the negative and positive tweets. Meanwhile, we can accept the null hypothesis and reject the corresponding alternative hypothesis on retweets, with no differences between the negative and positive tweets' impacts on the response rate of retweets.

Therefore, the differential effect of treatment shows the influence of sentiment polarity on the responses of Twitter within COVID-19 data. Sentiment polarity has a negative effect on the replies and favorites of the tweets. The negative topics on COVID-19 have more replies and favorites than those positive topics for both organizational and media tweets. However,

there isn't any significant difference between the retweets of both negative and positive topics on COVID-19.

## 5. Discussion

### 5.1 Principal results

With the data of 19,918 tweets on COVID-19, this study modeled the impact of the tweet's sentiment polarity (SP) on the responses, including replies, favorites, and retweets. The main results are summarized as follows.

First, this study's results support the argument that negative sentiment polarity attracts more replies (RP) and favorites (FV) than positive polarity but has no significant effect on retweets (RT). The negative topics on COVID-19 have more replies (69.95±49.01) and favorites (688.46±677.05) than the positive ones for the tweets. Li and Rao [5] suggest noticeable spikes in Twitter usage during disasters and other large events. Our findings also assist in understanding social media's mechanism of using negative sentiment to spread emotions in a compelling, effective, and fast way.

Second, the t-test results confirm the above results, with the difference between negative and positive sentiments being significant for replies (p = 0.001) and favorites (p = 0.029), but not retweets (p = 0.405). The existing evidence [16] suggests that the correlations between (positive) sentiment polarity of tweets and their responses are statistically significant. Our results provided empirical evidence (Fig 6) about the effect of the text's sentiment polarity on the responses they received on the Twitter platform.

Third, this study implemented the inverse probability weighting method to obtain the covariate-adjusted means. This approach reduced the bias of the treatment effect. The weighting results are -69.95 (±49.01), -688.46 (±677.05), and -325.20 (±416.25) for replies, retweets, and favorites, respectively. The result (-69.95) suggests that the treatment effect of the sentiment polarity on the negative group responses is higher than in the positive group. Comarela et al. [18] suggest that the covariates increase the fraction of replied or retweeted messages, while these covariates are balanced in the inverse probability weighting model. The covariate balance reduces the bias for identifying the treatment effect of sentiment polarity.

Moreover, this study applied a differential effect of treatment inference methodology to analyze social media data, which shows the potential strengths of randomized inference with multiple social media responses. The previous studies applied machine mining [16] and network analysis techniques [39] to extract hidden topics or examine the conversation stimulation mechanism on social media. Our study developed methods to analyze sentiment polarity impact on their responses on Twitter (Fig 1). This study provides quantitative evidence about the differential effect of sentiment polarity on the response. The results confirmed that the sentiment polarity with negative values would increase their responses on Twitter. Our study generates causal inference to support the argument that there is an increment of the replies and favorites in information sharing for the negative sentiment polarity, but no significant evidence with retweets.

### 5.2 Limitations

Our study limitations lie in the following aspects. First, all data on sentiment analysis and responses were collected from social media (the *Twitter* platform). The sample data were collected from a limited number of Twitter accounts. They were selected as the representative accounts because they had an enormous number of followers in the four counties (Australia, China, the UK, and the US). Despite separating the weekly data by every Sunday, the tweet

replies may also include the replies for other weekly data subsets. Therefore, a selection bias may exist in the measurement.

Second, this study considered the organizations and news agencies at the block level. Although the blocking factor increased the goodness of fit, it did not impact the treatment effect of randomized inference. The identification of the block effect (of different regions, organizations, or countries) is beyond this study's scope. Nevertheless, this blocking factor did not change the results of the treatment effect. The reason lies in the covariate balance. The bias due to the covariates is reduced by the weighting method when the negative and positive groups' entries overlap. The time factor is orthogonal to the treatment factor; thus, this study did not analyze the impact of time on the responses.

Third, the covariates contain the essential variables while omitting the others. Our study should be regarded as a starting point for further investigation rather than examining the determinants of response rates of tweets or final causal statements about the influence of sentiment polarity on tweets' response rates. Other determinants may also affect the number of response rates of tweets (e.g., the tweets' political attitudes). Relevant data can be collected to reduce the bias of results in future works. The logistical regression is only employed to obtain the balancing score (probability) and weights; thus, the regression coefficients are no longer useful parameters and are not reported.

The drawback of sentiment analysis is that the model is only focused on the terms but omits the meaning of an entire tweet. Additional results can be achieved through further exploration of more topics and Twitter accounts.

## 6. Conclusions

Social media is one of the essential channels for people to receive news. Different from the traditional media, on social media, multiple organizations can disclose different sentiments on the same event; and the readers can rapidly respond to these tweets. Its convenience and interactivity have seen social media, i.e., Twitter becomes one of the essential, shared resources for disclosing and tracking the trend of the COVID-19 pandemic. This has seen social media (i.e., Twitter) increasingly become a public channel for understanding people's feedback on current affairs with different narratives. It is also essential to study the readers' emotions along with the COVID-19 trend. This study provides a systematic way to understand people's reactions to negative and positive reports of the COVID-19 pandemic. The empirical findings prove that "Good things don't go out, bad things spread for thousands of miles".

The multiple responses (replies, favorites, and retweets) reflected the followers' spontaneous response in reading the tweets with different sentiment polarity. The results also provide new insights into the intersection of social media and public health outbreak surveillance. The Twitter data mining provided empirical evidence on the public organizations' engagement during the COVID-19 pandemic. This method is also useful to inform public health education on social media and compare the interactions of those public organizations on their communications channels to the public through social media. This study examined the tweets' responses within the COVID-19 topic with sentiment polarity. It contributes new insight for understating the influence of sentiment polarity on the tweets' responses on COVID-19. To reduce the selection bias, we will collect more relevant data, analyze and compare the topics to further understand social media dynamics in the future.

## Supporting information

**S1 File.**
(ZIP)

## Author Contributions

**Conceptualization:** Haiyan Yu, Ching-Chi Yang, Ping Yu.

**Data curation:** Haiyan Yu, Ke Liu.

**Formal analysis:** Haiyan Yu, Ke Liu.

**Investigation:** Haiyan Yu.

**Methodology:** Haiyan Yu, Ching-Chi Yang, Ke Liu.

**Software:** Haiyan Yu.

**Validation:** Haiyan Yu.

**Visualization:** Haiyan Yu.

**Writing – original draft:** Haiyan Yu, Ching-Chi Yang, Ping Yu.

**Writing – review & editing:** Haiyan Yu, Ching-Chi Yang, Ping Yu.

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
