## [Decision Letter · Decision Letter 0]

26 Jan 2022

PONE-D-21-40794Emotion diffusion effect: negative sentiment COVID-19 tweets of public organizations attract more responses from followersPLOS ONE

Dear Dr. Yang,

Thank you for submitting your manuscript to PLOS ONE. After careful consideration, we feel that it has merit but does not fully meet PLOS ONE’s publication criteria as it currently stands. Therefore, we invite you to submit a revised version of the manuscript that addresses the points raised during the review process.

We look forward to receiving your revised manuscript.

Kind regards,

Sanjay Kumar Singh Patel, Ph.D.

Academic Editor

PLOS ONE

Journal Requirements:

3. In your Methods section, please include additional information about your dataset and ensure that you have included a statement specifying whether the collection and analysis method complied with the terms and conditions for the source of the data.

Reviewers' comments:

Reviewer's Responses to Questions

**Comments to the Author**

1. Is the manuscript technically sound, and do the data support the conclusions?

Reviewer #1: Yes

Reviewer #2: Yes

Reviewer #3: Yes

2. Has the statistical analysis been performed appropriately and rigorously? 

Reviewer #1: Yes

Reviewer #2: Yes

Reviewer #3: Yes

3. Have the authors made all data underlying the findings in their manuscript fully available?

Reviewer #1: Yes

Reviewer #2: Yes

Reviewer #3: Yes

4. Is the manuscript presented in an intelligible fashion and written in standard English?

Reviewer #1: Yes

Reviewer #2: Yes

Reviewer #3: Yes

5. Review Comments to the Author

Reviewer #1: In this paper entitled "Emotion diffusion effect: negative sentiment COVID-19 tweets of public organizations attract more responses from followers", the authors investigated a new method to unveil the causal relationships between the sentiment clarity and responses in social media data. The sentiment polarity as the treatment variable is applied in the manuscript. The results indicated no significant difference in the number of retweets between negative and positive tweets. Still, negative sentiment polarity on COVID-19 attracted significantly more replies and favorites than the positive tweets. The manuscript is well written and easy to understand. The work done by the authors is fantastic and has proper statistical rigor. Therefore, there is no technical reason to reject this manuscript. I congratulate the author for this work.

Reviewer #2: To Author –

The manuscript titled “Emotion diffusion effect: negative sentiment COVID-19 tweets of public organizations attract more responses from followers” was quite interesting, and overall content of the manuscript has written very well. While appreciating their efforts in carrying out the work on sentiment polarity, there are many flaws in the manuscript, which needs to be addressed, before being accepted for publications.

Comments for revision:

1. The data collected from nine authoritative public organization from Dec. 01, 2019, to May 10, 2020. Could you give strong evidence to select the data only for the 24 Weeks?

2. What is the purpose of including the news agencies along with Govt. organization? Any reason behind it?

3. Some sentences are much lengthy in text and makes complicated in understanding. Try to avoid if possible.

4. Please explain how this study could fill the knowledge gap in the field and what the novelty it offers.

5. There are some related works published to understand public sentiment for the COVID‐19 outbreak, but some facts are not detailed properly in the manuscript –

a. How could these tweets make an impact on public health and their emotions?

6. Although various methods were developed previously, how these methods reflect the real-world actions is still uncertain. This may need to be discussed. Also, discuss about the future challenges with COVID-19 variants issues i.e. https://doi.ord/10.1007/s15010-021-01734-2.   

Reviewer #3: Summary:

The authors report a study titled “Emotion diffusion effect: negative sentiment COVID-19 tweets of public organizations attract more responses from followers”. The advent of the COVID19 pandemic study has led to several debates in the social media. In this study the authors analyzed data to social media giant Twitter. Literature suggests researchers were divided into two different thought camps. One group suggests positive sentiment polarity enhances a tweet's responses whereas the other suggest negative sentiment polarity enhances a tweet's responses. Through a meticulous study of large number of tweet’s, the current study supports the argument that negative sentiment polarity attracts more replies (RP) and favorites (FV) than positive polarity but has no significant effect on retweets (RT). Although the study is limited by the number of twitter accounts studied however these accounts has large number of followers providing a good platform of study.

Minor comment

Can you provide a reference for “To date, few studies have examined the impact of a tweet's sentiment polarity (SP) on its responses (i.e., replies, favorites, and retweets).”

---

## [Author Response · Author response to Decision Letter 0]

6 Feb 2022

RESPONSE to REVIEWERS' COMMENTS

REFERENCE: MS. REF. NO.: PONE-D-21-40794 

TITLE: EMOTION DIFFUSION EFFECT: NEGATIVE SENTIMENT COVID-19 TWEETS OF PUBLIC ORGANIZATIONS ATTRACT MORE RESPONSES FROM FOLLOWERS 

Dear Dr. Sanjay Kumar Singh Patel,

We would like to thank you for providing us with the opportunity to revise and resubmit our manuscript. We found your comments and suggestions constructive and valuable and did our best to address them. In this response document, we have provided point-by-point responses to your and reviewers' (#1 to 3) comments and suggestions. 

In the following sections, the reviewers' comments/suggestions are presented in italic font, whereas our responses to these comments/suggestions are in normal font. In the revised manuscript, major additions and changes are also edited 'with Track Changes'. Moreover, we also provided a separate file labeled 'Manuscript'.

RESPONSE TO REVIEWER #1'S COMMENTS

Specific comments

1. In this paper entitled "Emotion diffusion effect: negative sentiment COVID-19 tweets of public organizations attract more responses from followers", the authors investigated a new method to unveil the causal relationships between the sentiment clarity and responses in social media data. The sentiment polarity as the treatment variable is applied in the manuscript. The results indicated no significant difference in the number of retweets between negative and positive tweets. Still, negative sentiment polarity on COVID-19 attracted significantly more replies and favorites than the positive tweets. The manuscript is well written and easy to understand. The work done by the authors is fantastic and has proper statistical rigor. Therefore, there is no technical reason to reject this manuscript. I congratulate the author for this work.

Response. Thanks for your thorough reading and valuable comments. 

 

RESPONSE TO REVIEWER #2'S COMMENTS

Specific comments

The manuscript titled “Emotion diffusion effect: negative sentiment COVID-19 tweets of public organizations attract more responses from followers” was quite interesting, and overall content of the manuscript has written very well. While appreciating their efforts in carrying out the work on sentiment polarity, there are many flaws in the manuscript, which needs to be addressed, before being accepted for publications.

Comments for revision:

1. The data collected from nine authoritative public organization from Dec. 01, 2019, to May 10, 2020. Could you give strong evidence to select the data only for the 24 Weeks?

Response. Thanks for your thorough reading and valuable suggestions. This paper has been revised in strict accordance with your constructive suggestions and recommendations. As described in the paper, 

[Page 7] "Although published research [1] set up a comprehensive timeline of the spreading of COVID-19, we considered it useful to start initial data collection one week before the first reported case in Wuhan to set up the pre-COVID benchmark. The data collection was concluded at the week including May 10, 2020, to avoid the impact of the Black Lives Matter (BLM) movement [2] (see Figure 2). "

A. No. of COVID-19 confirmed case B. No. of COVID-19 deaths

Figure 2. Weekly number of COVID-19 confirmed cases and deaths.

 

2. What is the purpose of including the news agencies along with Govt. organization? Any reason behind it?

Response. Thanks for your valuable comments. The purpose of including the news agencies along with Govt. organization lies in two aspects. 

As described in the paper, 

[Page 5] "The reasons for including the news agencies along with government organizations lie in two aspects. First, news agencies were often employed in the literature to investigate the social media impact on an emergent event, e.g., the use of social media in the Westgate mall terror attack in Kenya [3]. Second, this paper employed the news agencies that are the representative media account on the Twitter platform from the four countries. These news agencies are one of the main communication channels of the stakeholders (i.e., governments, non-profit organizations, and healthcare communities that have taken active actions to respond to the COVID-19 epidemic) to the public."

3. Some sentences are much lengthy in text and makes complicated in understanding. Try to avoid if possible.

Response. Thanks for your thorough reading and constructive suggestions. This paper has been revised in strict accordance with your recommendations. The long sentences over 20 words in the paper have been mostly revised with many short sentences. For example, the sentence “Coronavirus disease 2019 (COVID-19) has triggered an enormous number of discussion topics on social media Twitter, with an impact on the global health system and citizen responses to the pandemic.” was replaced by two short sentences, “Coronavirus disease 2019 (COVID-19) has triggered an enormous number of discussion topics on social media Twitter. It has an impact on the global health system and citizen responses to the pandemic.”

4. Please explain how this study could fill the knowledge gap in the field and what the novelty it offers.

Response. Thanks for your thorough reading. This study filled the knowledge gap in the field as described in the abstract. 

[Page 1] "To date, studies on Twitter data have focused on the associational relationships between variables in a population. There is a need for further discovery of causality, such as the influence of sentiment polarity of tweet response on further discussion topics, which these topics often may reflect the human perception of COVID-19. This study addresses this exact topic. It aims to develop a new method to unveil the causal relationships between the sentiment polarity and responses in social media data."

Second, the novelty it offers is also described in the abstract. 

[Page 1] " We employed sentiment polarity, i.e., positive or negative sentiment, as the treatment variable in this quasi-experimental study…. Employing the inverse probability weighting model, we identified the treatment effect of sentiment polarity on the multiple responses of tweets. The topics with negative sentiment polarity on COVID-19 attracted significantly more replies (69±49) and favorites (688±677) than the positive tweets. However, no significant difference in the number of retweets was found between the negative and positive tweets. This study contributes a new method for social media analysis. It generates new insight into the influence of sentiment polarity of tweets about COVID-19 on tweet responses. "

5. There are some related works published to understand public sentiment for the COVID‐19 outbreak, but some facts are not detailed properly in the manuscript –

a. How could these tweets make an impact on public health and their emotions?

Response. Thanks for your constructive suggestions. This paper has been revised in strict accordance with your constructive suggestions. We have added more related works and made a thorough review to better understand the public sentiment on the COVID‐19. 

As described in the paper. 

[Page 3] "A high number of responses indicates a favorable effect of disseminating the message of the original tweet; therefore, understanding responses to a tweet has been a major topic of sentiment analysis. For example, cross-cultural polarity on COVID-19 related tweets was examined with emotion detection methods [4]. To date, few studies have examined the impact of a tweet's sentiment polarity (SP) on its responses (i.e., replies, favorites, and retweets). This study aims to fill this gap."

[Page 5] "The types of health topics discussed on Twitter were investigated in [5] and they found that tweets can both augment existing public health capabilities and enable new ones. Through the analysis of a unique Twitter dataset captured in the early stage of the current Ebola outbreak, the results provided insight into the intersection of social media and public health outbreak surveillance [6]. Their study suggests that Twitter mining is useful to inform public health education. Moreover, a systematic review also suggests that there is an essential need for an accurate and tested tool for sentiment analysis of tweets using a health care setting [7]. For example, the natural language processing approach and healthcare-specific corpus of manually annotated tweets were implemented to learn the sentiment from Texas Public Agencies’ tweets and public engagement during the COVID-19 pandemic [8]."

6. Although various methods were developed previously, how these methods reflect the real-world actions is still uncertain. This may need to be discussed. Also, discuss about the future challenges with COVID-19 variants issues i.e. https://doi.ord/10.1007/s15010-021-01734-2. 

Response. Thanks for your thorough valuable suggestion and recommendations. We rewrote the discussion section with more details on the impacts and potential outcomes of the real-world action and the future challenges with COVID-19 variants issues [9].

As described in the paper. 

[Page 13] "The multiple responses (replies, favorites, and retweets) reflected the followers' spontaneous response in reading the tweets with different sentiment polarity. 

The results also provide new insights into the intersection of social media and public health outbreak surveillance. The Twitter data mining provided empirical evidence on the public organizations' engagement during the COVID-19 pandemic. This method is also useful to inform public health education on social media and compare the interactions of those public organizations on their communications channels to the public through social media. This study examined the tweets' responses within the COVID-19 topic with sentiment polarity. It contributes new insight for understating the influence of sentiment polarity on the tweets' responses on COVID-19. To reduce the selection bias, we will collect more relevant data, analyze and compare the topics to further understand social media dynamics in the future. "

RESPONSE TO REVIEWER #3'S COMMENTS

Specific comments

The authors report a study titled “Emotion diffusion effect: negative sentiment COVID-19 tweets of public organizations attract more responses from followers”. The advent of the COVID19 pandemic study has led to several debates in social media. In this study, the authors analyzed data to social media giant Twitter. Literature suggests researchers were divided into two different thought camps. One group suggests positive sentiment polarity enhances a tweet's responses whereas the other suggest negative sentiment polarity enhances a tweet's responses. Through a meticulous study of large number of tweet’s, the current study supports the argument that negative sentiment polarity attracts more replies (RP) and favorites (FV) than positive polarity but has no significant effect on retweets (RT). Although the study is limited by the number of twitter accounts studied however these accounts has large number of followers providing a good platform of study.

Response. Thanks for your thorough reading and valuable comments. 

2. Minor comment: Can you provide a reference for “To date, few studies have examined the impact of a tweet's sentiment polarity (SP) on its responses (i.e., replies, favorites, and retweets).”

Response. Thanks for the constructive suggestions. This paper has been revised in strict accordance with your constructive suggestions and recommendations. The following shows a reference that examined the tweet's sentiment polarity [4], but few studies have examined its impact on the responses of a tweet. 

As described in the paper, 

[Page 3] "For example, cross-cultural polarity on COVID-19 related tweets were examined with emotion detection methods [4]. To date, few studies have examined the impact of a tweet's sentiment polarity (SP) on its responses (i.e., replies, favorites, and retweets). "

REFERENCES

1. Secon H, Woodward A, Mosher D. A comprehensive timeline of the coronavirus pandemic at 6 months, from China's first case to the present https://www.businessinsider.com/coronavirus-pandemic-timeline-history-major-events-2020-3: business insider; Jun 30, 2020 [September 17, 2020]. Available from: https://www.businessinsider.com/coronavirus-pandemic-timeline-history-major-events-2020-3.

2. Bejan V, Hickman M, Parkin WS, Pozo VF. Primed for death: Law enforcement-citizen homicides, social media, and retaliatory violence. PloS one. 2018;13(1):e0190571.

3. Simon T, Goldberg A, Aharonson-Daniel L, Leykin D, Adini B. Twitter in the cross fire—the use of social media in the Westgate Mall terror attack in Kenya. PloS one. 2014;9(8):e104136.

4. Imran AS, Daudpota SM, Kastrati Z, Batra R. Cross-cultural polarity and emotion detection using sentiment analysis and deep learning on COVID-19 related tweets. IEEE Access. 2020;8:181074-90.

5. Dredze M. How social media will change public health. IEEE intelligent systems. 2012;27(4):81-4.

6. Odlum M, Yoon S. What can we learn about the Ebola outbreak from tweets? American journal of infection control. 2015;43(6):563-71.

7. Gohil S, Vuik S, Darzi A. Sentiment analysis of health care tweets: review of the methods used. JMIR public health and surveillance. 2018;4(2):e5789.

8. Tang L, Liu W, Thomas B, Tran HTN, Zou W, Zhang X, et al. Texas public agencies’ tweets and public engagement during the COVID-19 pandemic: Natural language processing approach. JMIR public health and surveillance. 2021;7(4):e26720.

9. Thakur V, Bhola S, Thakur P, Patel SKS, Kulshrestha S, Ratho RK, et al. Waves and variants of SARS-CoV-2: understanding the causes and effect of the COVID-19 catastrophe. Infection. 2021:1-16.

---

## [Decision Letter · Decision Letter 1]

17 Feb 2022

Emotion diffusion effect: negative sentiment COVID-19 tweets of public organizations attract more responses from followers

PONE-D-21-40794R1

Dear Dr. Yang,

We’re pleased to inform you that your manuscript has been judged scientifically suitable for publication and will be formally accepted for publication once it meets all outstanding technical requirements.

Kind regards,

Sanjay Kumar Singh Patel, Ph.D.

Academic Editor

PLOS ONE

Reviewers' comments:

Reviewer's Responses to Questions

**Comments to the Author**

1. If the authors have adequately addressed your comments raised in a previous round of review and you feel that this manuscript is now acceptable for publication, you may indicate that here to bypass the “Comments to the Author” section, enter your conflict of interest statement in the “Confidential to Editor” section, and submit your "Accept" recommendation.

Reviewer #2: All comments have been addressed

Reviewer #3: All comments have been addressed

2. Is the manuscript technically sound, and do the data support the conclusions?

Reviewer #2: Yes

Reviewer #3: Yes

3. Has the statistical analysis been performed appropriately and rigorously? 

Reviewer #2: Yes

Reviewer #3: Yes

4. Have the authors made all data underlying the findings in their manuscript fully available?

Reviewer #2: Yes

Reviewer #3: Yes

5. Is the manuscript presented in an intelligible fashion and written in standard English?

Reviewer #2: Yes

Reviewer #3: Yes

6. Review Comments to the Author

Reviewer #2: To Author –

The authors have made changes to the manuscript in response to the comments. As a result, there are no technical reasons to reject this entry. I congratulate the author on this achievement.

---

## [Editor Report · Acceptance letter]

28 Feb 2022

PONE-D-21-40794R1 

Emotion diffusion effect: negative sentiment COVID-19 tweets of public organizations attract more responses from followers 

Dear Dr. Yang:

I'm pleased to inform you that your manuscript has been deemed suitable for publication in PLOS ONE. Congratulations! Your manuscript is now with our production department. 

Kind regards, 

on behalf of

Dr. Sanjay Kumar Singh Patel 

Academic Editor

PLOS ONE